# Feeding Dairy Ewes with Fresh or Dehydrated Sulla (*Sulla coronarium* L.) Forage. 2. Effects on Cheese Enrichment in Bioactive Molecules

**DOI:** 10.3390/ani12182462

**Published:** 2022-09-18

**Authors:** Marialetizia Ponte, Giuseppe Maniaci, Antonino Di Grigoli, Riccardo Gannuscio, Mansour Rabie Ashkezary, Margherita Addis, Marianna Pipi, Marco Alabiso, Massimo Todaro, Adriana Bonanno

**Affiliations:** 1Dipartimento Scienze Agrarie, Alimentari e Forestali (SAAF), Università degli Studi di Palermo (UNIPA), 90128 Palermo, Italy; 2AGRIS Sardegna, Olmedo, 07040 Sassari, Italy

**Keywords:** dehydrated forage, condensed tannins, sheep, cheese, polyphenols, vitamins, fatty acids, health properties, biomarkers of feeding regime

## Abstract

**Simple Summary:**

Feeding ruminants with fresh sulla forage, containing polyphenols with antioxidant activity, seems to improve the performance of animals and the technological, nutritional and health properties of dairy products. However, the potential of sulla forage can be compromised by traditional methods, such as hay-making, to produce conserved forage; alternatively, dehydration could be an appropriate solution to preserve the content in bioactive components. This paper reports the effects of feeding dehydrated or fresh sulla forage on physico-chemical traits and health properties, due to antioxidant activity and fatty acid profile, of sheep cheese. An attempt was made also to identify cheese constituents that, being able to discriminate the ewes’ diet, may be proposed as biomarkers to trace cheese produced from animals fed with fresh forage. The results showed that the use of fresh or dehydrated sulla was able to enhance the health properties of cheeses, so dehydration can represent an opportunity to preserve the potential of fresh sulla forage. Moreover, α-linolenic acid and its ratio with linoleic acid showed the ability to discriminate cheese in relation to the animals’ feeding regime.

**Abstract:**

Sulla is a tanniferous legume species largely used as fresh or preserved forage in the ruminants’ diets. Due to its content in polyphenols with antioxidant activity, especially condensed tannins (CT), fresh sulla forage (FSF), when eaten by ruminants, is able to enrich animal products with antioxidant molecules and polyunsaturated fatty acids (PUFA) that are beneficial for consumers’ health. Dehydration represents a valid alternative to haymaking to preserve these properties also in periods when FSF is unavailable. In this research, the effects of ewes’ diets based on sulla hay (SH), pelleted dehydrated sulla forage (DSF) or FSF were evaluated on cheese physico-chemical traits, including fatty acid (FA) profile, vitamins A and E, polyphenols, antioxidant capacity and oxidative stability. The individual daily milk from 10 first lambing (FL) and 10 third lambing (TL) Valle del Belìce ewes at about 60 days in milk, divided homogeneously into 5 groups fed different diets in a partial 5 × 2 Latin square design with 2 phases, and the bulk milk of farming ewes fed only on natural pasture, were used in 2 sessions of micro cheese-making to manufacture cheeses, sampled at 48 h of ripening. The experimental diets were: SHL = SH ad libitum; DSF2 = 2 kg/day DSF per head plus SH ad libitum; FSF2 = 2 kg/day FSF per head plus SH ad libitum; FSF4 = 4 kg/day FSF per head plus SH ad libitum; FSFL = FSF ad libitum. Concentrate was supplied at the amount of 0.8 or 1.2 kg/day per head for FL and TL, respectively. Cheese health properties greatly improved with FSFL diet, based on the exclusive use of fresh forage, that induced the increase in the content of molecules with antioxidant activity, as vitamin A, vitamin E and polyphenols, enhancing the oxidative stability, and the level of PUFA as rumenic acid (C18:2 c9t11) and α-linolenic acid (ALA, C18:3n-3). Cheeses from the DSF2 group showed levels of vitamin A, vitamin E and PUFA higher than SHL cheeses and comparable to those of FSF4 cheeses, whereas their ALA content was not different from that of FSFL cheeses. Evaluating those molecules affected by the level of fresh forage in the diet for their ability to trace the animals’ feeding regime, ALA and its ratio with linoleic acid (LA, C18:2n-6) (LA/ALA) exhibited an effective role as biomarker in discriminating cheese from animals fed fresh forage-based diets. The results showed how dehydrated sulla pellets, as an alternative to hay in periods of limited availability of fresh forage, can preserve the nutritional and health properties of dairy products with regards to their antioxidant capacity and FA profile.

## 1. Introduction

The growing interest in the use of plant polyphenols in ruminants’ nutrition is strictly linked to the biological activities that these natural molecules exert in the rumen, as well as to their antioxidant activity, with beneficial implications for environment, animal and human health and food quality [1,2,3,4].

In ruminants feeding, the presence of polyphenols showed to optimize rumen fermentation, contributing to reducing methanogenesis and mitigating methane emission into the environment [3,5]; to reduce the dietary protein degradation in the rumen, favouring a greater availability of aminoacids for intestinal absorption [1,5,6] and the reduction of nitrogen excretions into the environment [7]; to protect the polyunsaturated fatty acids (PUFA) from the rumen biohydrogenation, facilitating the transfer in milk and meat and derived products, thus enhancing their nutritional and health value [8]; to improve the antioxidant status and immune defenses of animals, ensuring their welfare and health, and the oxidative stability of their products [4,9,10].

On this basis, the intake of polyphenol-rich plants by ruminants may be certainly a promising strategy for improving the nutritional value, health properties and shelf life of ruminant-derived foods. Among forage species, sulla (*Sulla coronarium* L.), a biennial legume widely used in Mediterranean areas for grazing or silage and hay production [11], represents an excellent source of polyphenols, consisting mainly of proanthocyanidins, known as condensed tannins (CT) [5,12,13], present at moderate concentrations (<60 g/kg DM) [14,15]. Indeed, the positive impact of sulla forage on ruminants’ performance and the quality of their products is mainly attributed to its moderate CT content. In this regard, ruminants fed with sulla forage showed to enhance the efficiency of feed utilization and dairy performance [7,16,17,18], as well as their plasma oxidative status [9]. The moderate levels of CT in sulla forage were also able to inhibit the complete biohydrogenation of dietary PUFA in the rumen, thus enriching dairy products [17,18,19] and meat [20,21] with PUFA beneficial for human health.

Furthermore, the intake of CT seemed to be responsible for the transfer of phenolic compounds to various tissues [4,22] thus contributing to increase the antioxidant capacity and the oxidative stability of dairy products [9].

In this context, the use of diets based on sulla forage seems to be promising to improve the performance of ruminants and especially the technological, nutritional and health properties of dairy products, as well as their oxidative stability. However, the potential of sulla forage containing CT can be strongly compromised by the processes to produce preserved forage for the periods of low availability of fresh forage. As a consequence, the introduction of sulla in forage systems implies the necessity to improve the drying techniques to produce preserved forages, stabilizing their content in nutrients and bioactive molecules that can be transferred to the animals’ products. In this regard, dehydration and pelleting could be suitable solutions to preserve and stabilize the content in nutrients and bioactive components of sulla forage, as an alternative to avoid their loss provoked by the hay-making process.

The positive effects that fresh forage-based diets induce on nutritional and health properties of ruminants’ products have increased the interest of consumers in animal-derived foods obtained in pasture-based systems, which ensure safe products together with respect for the environment and animal welfare [23,24]. Accordingly, benefits for both consumers and producers would derive from the individuation of specific biomarkers that can discriminate the products for their production system [25,26], especially to distinguish those obtained from animals fed fresh forage at pasture. In this regard, some components of lipid fraction seem to have the potential as effective biomarkers to trace cheese for animals’ feeding system. In particular, the determination of some PUFA as α-linolenic acid (ALA) and rumenic acid (RA, C18:2 c9t11), that is the main isomer of conjugated linoleic acids (CLA), as well as some branched-chain FA and vitamin A, showed to represent a promising approach for traceability of cheese production system [25,26,27].

Thus, this study was planned in order to verify the validity of dehydration in preserving those nutrients and bioactive molecules of fresh sulla forage, commonly lost with hay-making, and then ascertain the ability of dehydrated sulla forage in enhancing the tecnological, nutritional, and health properties of dairy products in periods of low availability of fresh forage.

The first part of this study [28] reports the results regarding the effects of feeding regimes based on fresh or dehydrated sulla forage on feed intake, digestibility, milk production and oxidative status of dairy ewes. This paper reports the effects of feeding pelleted dehydrated or fresh sulla forage at different levels on physico-chemical traits and health properties of sheep cheese, with particular attention to antioxidant activity and FA profile. A further object of this study was to identify cheese components able to discriminate the ewes’ diet, that may be proposed as molecular biomarkers for traceability of cheese produced from milk of animals fed with fresh forage-based diets. With this aim, the cheeses derived from ewes fed diets based on different types and levels of sulla forage were compared to those from ewes fed exclusively on natural pasture.

## 2. Materials and Methods

### 2.1. Animals, Experimental Design, and Feeding Treatments

This study was conducted in a commercial farm rearing ewes of Valle del Belìce breed, located in Sicily (Santa Margherita di Belìce, Agrigento, Italy; 37°41′ N; 13°00′ E; 411 m above sea level) for 6 weeks in the spring. A total of 20 ewes at about 60 days in milk were assigned homogeneously to 5 groups in relation to parity (10 first and 10 third lambing ewes), live weight (48.86 ± 5.85 kg) and milk yield (2101 ± 407 g/day) and housed in pairs into pens placed indoors. Each experimental group received one of 5 diets, according to a partial 5 × 2 Latin square design with 2 phases, each composed of 10 days for adaptation to diets and 5 days of experimental period. The diets, differing for the forage component, were as follows: SHL = sulla hay (SH; crude protein (CP) 7.49% DM; net energy for lactation (NE_L_) 695 kcal/kg DM) ad libitum; DSF2 = 2 kg/day dehydrated sulla forage (DSF, CP 14.72% DM; NE_L_ 933 kcal/kg DM) per head plus SH ad libitum; FSF2 = 2 kg/day fresh sulla forage (FSF, CP 14.82% DM; NE_L_ 1349 kcal/kg DM) per head plus SH ad libitum; FSF4 = 4 kg/day FSF per head plus SH ad libitum; FSFL = FSF ad libitum. Ewes were supplemented with a commercial concentrate feed (CCF, CP 17.71% DM; NE_L_ 1791 kcal/kg DM) offered at levels of 800 g/day per head to first lambing ewes (FL) and 1200 g/day per head to third lambing ewes (TL).

Forages provision and ewes’ management, in relation to housing and feeding, were performed as described by Gannuscio et al. [28]. The experiment protocol had the approval of the Animal Welfare Body of the University of Palermo (2021-UNPA-CLE-132953) who ruled as not applicable the requirements established by the National Legislative Decree n. 26/2014, implementing the Directive 2010/63/EU.

### 2.2. Milk Sampling and Cheese Manufacturing

On the last day of each of the 2 experimental phases, daily milk yield of each of the 20 ewes was recorded and the relative samples, ranging from 1 to 1.5 kg, were collected. Thus, a total of 40 individual milk samples were collected and processed in 2 sessions, one for each phase, developing a laboratory-scale micro cheese-making procedure to obtain individual pressed-curd type cheeses. In both micro cheese-making sessions, cheeses were manufactured also using the bulk milk of farming ewes fed exclusively on natural pasture.

The main laboratory equipment included a heater fitted with a thermostat and four water baths in which a uniform temperature was ensured by digital controllers and pumps for water mixing. Since every water bath contained 6 vats of 2-L capacity, the apparatus allowed processing the 20 samples simultaneously. Each refrigerated (4 °C) milk sample was placed in a pyrex glass beaker and heated in the water bath for about 30 min to reach 37 °C. After addition of 0.3 g/kg of lamb rennet paste (title 1:10,000) diluted in distilled water (7.5:100, *w*/*v*), milk was held at 37 °C for 1 h until coagulation. The curd was broken into small particles like rice grains using a glass stick, then cooked at 80 °C for 4 min in the water bath, removed from the beaker and manually pressed into a cylindrical, perforated plastic mould of 10 cm diameter to drain the whey, and turned every 3 min to facilitate draining. After about 15 min, each cheese within the mould was held in the water bath at 60 °C for 1 h, and then was placed on a flat surface for draining, weighed after 1 h, and transferred to a ripening room at a temperature of 16 °C and a relative humidity of 80%; after 48 h cheeses were weighted to calculate cheese yield and then sampled for analysis.

### 2.3. Milk and Cheeses Analyses

#### 2.3.1. Milk Physico-Chemical Traits

Individual milk samples were analysed for lactose, fat, urea and somatic cells count (SCC) by infrared method (Combi-foss 6000, Foss Electric, Hillerød, Denmark), total bacterial count (TBC) by BactoScan instrument (Foss Electric), pH by HI 9025 pH meter (Hanna Instruments, Ann Arbor, MI, USA), and titratable acidity by Soxhlet-Henkel method (°SH/50 mL). Total nitrogen (TN), non-casein nitrogen (NCN) and non-protein nitrogen (NPN) were determined by standard FIL-IDF procedures [29,30] to calculate total protein (TN × 6.38), casein [TN − (NCN × 0.994) × 6.38] and whey protein [(NCN − NPN) × 6.38). Clotting parameters such as coagulation time (r, min), curd-firming time (k_20_, min), curd firmness (a_30_, mm), and curd firmness after twice the clotting time (a_2r_, mm), were also measured in 10 mL of milk at 35°C added with 0.2 mL diluted solution (1.6:100, *v*/*v*) of rennet (1:15,000; Chr. Hansen, Parma, Italy) by a Formagraph instrument (Foss Electric).

#### 2.3.2. Cheese Physical Traits

Cheeses were sampled at 48 h and evaluated for their physical traits. Cheese colour was assessed in duplicate on internal and external surfaces measuring, by a Minolta Chroma Meter CR-300 (Minolta, Osaka, Japan), the values of lightness (L*, from 0 = black to 100 = white), redness (a*, from green = −a to red = +a) and yellowness (b*, from blue = −b to yellow = +b), according to the CIE L*a*b* system [31]. Cheeses hardness was evaluated measuring the maximum resistance to compression (compressive stress, N/mm^2^) of samples (2 × 2 × 2 cm) kept at room temperature (22°C), with an Instron 5564 tester (Instron, Trezzano sul Naviglio, Milano, Italy).

#### 2.3.3. Cheese Chemical Composition

The 48-h cheese samples were lyophilized for successive analyses. International Dairy Federation standards were performed to determine dry matter (DM) [32], fat [33], protein (N × 6.38) [34] and ash [35]. The α-tocopherol (vitamin E) and total retinol (vitamin A) in the cheese samples were determined in duplicate by reversed-phase HPLC methods, as reported by Panfili et al. [36] and Manzi et al. [37]. Briefly, aliquots of lyophilized cheese (0.15 g) were digested with 2 mL of KOH (60% aqueous solution, *w*/*v*), 2 mL of 95% ethanol, 1 mL of NaCl (1% aqueous solution, *w*/*v*), and 5 mL of an ethanolic solution of pyrogallol (6%, *w*/*v*) added as an antioxidant. After digestion in a water bath at 70 °C, the suspension was cooled for 30 min, added with 5 mL of a NaCl solution (1%, *w*/*v*) to prevent emulsification, and then extracted with 10 mL of n-hexane/ethyl acetate (9:1, *v*/*v*). The lower aqueous layer was extracted 3 more times, with 5 mL of n-hexane/ethyl acetate (9:1, *v*/*v*). The pooled organic layers were evaporated with a rotary evaporator at 30 °C, and the dry sample was dissolved in 3 mL of methanol for HPLC. A sample volume of 20 µL was injected into HPLC equipment, previously filtered using a 0.20 µm PTFE filter. A liquid chromatography Agilent Series 1100 (Agilent Technologies, Palo Alto, CA, USA) equipped with a Zorbax ODS column, 4.6 mm i.d. × 150 mm (Agilent Technologies, Palo Alto, CA, USA) and a programmable spectrofluorometer detectors, FLD detector Model 1260 Infinity G1321A (Agilent Technologies, Palo Alto, CA, USA), was used for all the HPLC analyses. The temperature of the column was kept at 25 °C, the flux of the mobile phase (methanol/H2O (98:2, *v*/*v*) isocratic elution) was set at 1 cm^3^/min. In the same chromatographic run Vitamin A (excitation 325 nm, emission 480 nm) and Vitamin E (excitation 293 nm, emission 326 nm) were detected. Chromatographic data were processed using the software Open Lab (Agilent Technologies, Palo Alto, CA, USA).

#### 2.3.4. Cheese Antioxidant Properties

Extracts of lyophilized cheese samples were prepared according to the method of Rashidinejad et al. [38] with slight modifications. Briefly, 0.5 g of milled cheese sample was suspended in 25 mL of methanol (95% aqueous solution) containing 1% HCl and homogenized for 30 sec by an Art-Miccra D-8 high-speed homogenizer (Art Labortechnik, Müllheim, Germany). The suspension was kept in an ultrasonic water bath (LBS1 Sonicator; Falc Instruments, Treviglio, Italy) at 40 °C for 30 min during which was vortexed for a few seconds every 10 min. Then it was cooled and filtered with cheesecloth, centrifuged at 7000 rpm for 10 min at 9 °C, and kept at −18 °C until analysis.

Extracted cheeses were analysed in duplicate for their antioxidant properties, measuring condensed tannins (CT), total phenolic compounds, and the trolox equivalent antioxidant capacity (TEAC). The content in CT was quantified by the butanol-HCl assay [39], using delphinidin as standard [40] and reading the absorbance at 550 nm in a HUCH DR3900 spectrophotometer (Hach, Loveland, CO, USA). Delphinidin aqueous solutions ranging from 0 to 0.15 mg/mL were used to obtain the calibration curve (R^2^ = 0.99). The results were expressed as delphinidin equivalent (g DE/kg DM).

The total polyphenols were determined by the Folin–Ciocalteau colorimetric method [41], with gallic acid as standard. Briefly, 100 µL sample extract were mixed with 900 µL distilled water and 500 µL Folin-Ciocalteau reagent diluted to a concentration of 1 N with distilled water. After the addition of a 2.5 mL of a 20% (*w*/*v*) sodium carbonate aqueous solution, the mixture was vortexed for 30 s and incubated in darkness at room temperature for 40 min. The absorbance of the samples was read at 725 nm in the HUCH DR3900 spectrophotometer. Gallic acid aqueous solutions of different concentrations (0–1 mg/mL) were used for the calibration curve (R^2^ = 0.99). The results were expressed as gallic acid equivalent (g GAE/kg DM).

The antioxidant activity of cheese extracts was evaluated by TEAC assay, according to Re et al. [42], as described by Bonanno et al. [43]. This is a decolorization assay measuring the radical scavenging ability of samples using the ABTS radical cation (ABTS•+), and trolox as standard. Briefly, the ABTS radical cation was produced by reacting to a 14 mM ABTS aqueous solution with an equal volume of 4.9 mM potassium persulfate and incubating the mixture in the dark at room temperature for 16 h before use. For the assay, the ABTS•+ solution was diluted with 5 mM phosphate-buffered saline (PBS) (pH 7.4) to an absorbance of 0.750 (±0.02) at 734 nm. The absorbance of a mixture of 40 µL of distilled water with 4 mL of a diluted ABTS•+ solution was read at 734 nm after incubation at 30 °C for 6 min. In the same way, 40 µL of each extracted sample was mixed with 4 mL diluted ABTS radical cation solution, and the absorbance read at 734 nm after incubation at 30 °C for 6 min was used to calculate the percentage decrease of the absorbance due to the decolorization in comparison with the absorbance read with distilled water. A trolox solution in PBS, ranging from 0 to 2.5 mM, was used to develop a calibration curve (R^2^ = 0.99), and the results were expressed as mmol trolox/kg DM of cheese.

#### 2.3.5. Cheese Oxidative Stability

The oxidative stability of cheese fat was assessed determining in duplicate the peroxide value (POV, mEq O_2_/kg fat) as index of primary lipid oxidation [44]. Moreover, the thiobarbituric acid–reactive substances (TBARS) as a measure of secondary lipid oxidation, expressed as µg malonylaldehyde (MDA/kg DM), was determined according to the methods proposed by Tarladgis et al. [45] and modified by Mele et al. [46]. Briefly, 4 g lyophilized cheese was mixed with 8 mL phosphate buffer aqueous solution (pH 7) and homogenized using an Art-Miccra D-8 high-speed homogenizer (Art Labortechnik). After addition of 30% (*v*/*v*) trichloroacetic acid aqueous solution (2 mL), the sample was vortexed for a few seconds, then filtered with Whatman No. 1 filter paper. An aqueous solution of 0.02 M thiobarbituric acid (5 mL) was added to 5 mL filtrate, and the mixture was placed in a hot water bath (90 °C) for 20 min then refrigerated. After being centrifuged at 4500 rpm for 5 min, the absorbance of the supernatant was read at 530 nm using the Hach DR/4000 U spectrophotometer. To quantify TBARS, 1,1,3,3-tetramethoxypropane solutions at concentrations ranging from 0.016 to 0.165 µg/mL were used for the calibration curve (R^2^ = 0.99).

#### 2.3.6. Cheese Fatty Acid Profile

Fatty acids (FA) in lyophilized cheese samples (100 mg) were directly methylated in 1 mL hexane with 2 mL 0.5 M NaOCH3 at 50 °C for 15 min, followed by 1 mL 5% HCl in methanol at 50 °C for 15 min, based on the bimethylation procedure described by Lee and Tweed [47]. Fatty acid methyl esters (FAME) were recovered in 1.5 mL hexane. Using an autosampler, 1 μL of each sample was injected into an HP 6890 gas chromatography system equipped with a flame-ionisation detector (Agilent Technologies, Santa Clara, CA, USA). FAME from each sample were separated using a CP-Sil 88 capillary column (100 m long, 0.25 mm internal diameter, 0.25 µm film thickness) (Chrompack, Middelburg, The Netherlands). The injector temperature was kept at 255 °C and the detector temperature was kept at 250 °C, with hydrogen flow of 40 mL/min, air flow of 400 mL/min, and a constant helium flow of 45 mL/min. The initial oven temperature was held at 70 °C for 1 min, increased by 5 °C/min to 100 °C, held for 2 min, increased by 10 °C/min to 175 °C, held for 40 min, then finally increased by 5 °C/min to a final temperature of 225 °C held for 45 min. Helium, with a pressure of 158.6 kPa and a flow rate of 0.7 mL/min (linear velocity 14 cm/s), was used as carrier gas. A FAME hexane mix solution (Nu-Check-Prep, Elysian, MN, USA) was used to identify each FA. Individual standards (Larodan Fine Chemicals AB, Malmö, Sweden) were used to identify some branched FA, as C15:0 iso, C15:0 anteiso, C17:0 iso, and C17:0 anteiso. Isomers of CLA were identified by using a standard mixture of C18:2 c9 t11 (rumenic acid, RA) and C18:2 c10 t12 methyl esters (Sigma-Aldrich, Milano, Italy) and published isomeric profiles [48,49]. To quantify total FA, C23:0 (Sigma-Aldrich) was used as internal standard (4 mg/g lyophilized cheese).

Based on FA profile, some indexes of health value of cheese fat were calculated as follows.

Thrombogenic index (TI) = (C14:0 + C16:0 + C18:0)/(0.5 × MUFA + 0.5 × n-6 PUFA + 3 × n-3 PUFA + n-3/n-6) [50].

Health-promoting index (HPI) = (n-3 PUFA + n-6 PUFA + MUFA)/(C12:0 + 4 × C14:0 + C16:0) [51].

Hypocholesterolemic FA to hypercholesterolemic FA ratio (h/H) = (C18:1 c9 + C18:2 n-6 + C20:4 n-6 + C18:3 n-3 + C20:5 n-3 + C22:5 n-3 + C22:6 n-3)/(C14:0 + 16:0) [52].

In addition, the General Health Index of Cheese (GHIC), based on the contents of total PUFA, n-3 PUFA and RA (CLA, C18:2 c9t11), together with polyphenols content and antioxidant capacity by TEAC, as proposed by Giorgio et al. [53], was applied as indicator of the health value of cheeses; the GHIC was calculated by adding the scores attributed to each of components according to a scale from 0 (minimum value) to 10 (maximum value), as described by Giorgio et al. [53], with the modifications to use the FA levels expressed as g/100 g FA, and the other components referred to cheese DM.

To express the efficiency of Δ9 -desaturase activity in the mammary gland, the desaturation indexes of C14:1, C16:1, C18:1 and C18:2 c9 t11 were calculated as follows.

C14:1 c9 Δ-9 desaturase ratio = C14:1 c9/C14:1 c9 +C14:0.

C16:1 c9 Δ-9 desaturase ratio = C16:1 c9/C16:1 c9 +C16:0.

C18:1 c9 Δ-9 desaturase ratio = C18:1 c9/C18:1 c9 +C18:0.

RA Δ-9 desaturase ratio = RA/trans vaccenic acid (TVA) + RA.

### 2.4. Statistical Analysis

Data of individual milk and cheese samples were analysed statistically using the MIXED procedure in SAS 9.2 software [54]. In the mixed model, experimental phase (2 levels), parity (*p*, 2 levels = TP and FL) and diet (D, 5 levels = SHL, DSF2, FSF2 FSF4 and FSFL) were fixed factors and the ewe was the random factor used as error term. All the possible interactions among fixed factors were tested and being always not significant, removed from the model. Before analysis, SCC and TBC values were transformed into logarithmic form (log_10_). When the effect of diet resulted significant (*p* ≤ 0.05), means were compared using *p*-values adjusted according to the Tukey-Kramer multiple comparison test. Using data of ewes’ feed intake reported by Gannuscio et al. [28] and cheese traits from SHL, FSF2, FSF4, FSFL experimental groups and from ewes fed pasture-based diet, linear and quadratic regressions were performed to identify the cheese parameters better related (R^2^ > 0.50) to ewes’ fresh forage DM intake to be proposed as biomarkers to differentiate cheeses for the animals’ feeding system.

## 3. Results and Discussion

### 3.1. Milk Physico-Chemical Traits

Physico-chemical parameters of individual milk used for micro cheese-making are reported in Table 1. On the whole, yield, composition and clotting ability of processed milk correspond to those of milk production monitored during the entire experiment and reported by Gannuscio et al. [28]. Milk yield obtained with FSFL and DSF2 diets was analogous and higher by about 150 g/day than those from ewes fed the other diets, which did not differ among them. Regarding milk components, only casein was significantly affected by diet; indeed, casein in FSFL milk was higher than that in milk from ewes fed FSF2 and SHL diets containing the lowest or no amount of FSF, whereas was comparable to that of DSF2 and FSF4 milk. The higher casein content of FSFL milk can explain the higher pH value, whereas did not contribute to improving milk clotting ability since no significant difference was observed in clotting parameters among all milk samples, except for the curd firmness (a_2r_, mm) which was the lowest in SHL milk, denoting a certain worsening in its clotting ability.

A significant effect of diet also emerged for the somatic cells count which was higher in milk from FSF4 group, due to a unique ewe showing a high count without visible signs of mastitis. Whereas, the level of total bacteria count did not differ among diets, and in general indicated a good hygienic quality of milk.

Regarding the effect of parity, as expected ewes at third lambing produced more milk than those at first lambing (about +450 g/day) with a lower protein content [55].

These results showed once again as feeding FSF containing CT favours milk yield and milk casein synthesis, in accordance with previous investigations [7,17,18], and evidenced also as DSF improved milk yield similarly to FSF without substantial changes in milk composition.

### 3.2. Cheese Physico-Chemical Traits

Table 2 shows the physico-chemical parameters of cheeses manufactured from individual milk. Cheese yield tended to be lower with milk from FSFL ewes, despite its higher casein content. This result can be attributed in part to the higher milk yield with FSFL diet that, in comparison with the other diets, contributed to the reduction of milk fat (4.63%, Table 1), although at a non-significant level; in this regard, the superior casein content of FSFL (4.03%, Table 1) was not able to balance the lower fat. However, also the higher water loss, indicated by the higher DM level recorded at a non- significant level in FSFL cheeses (54.72%), could have contributed to reducing their cheese yield. Nevertheless, the higher protein content of FSFL cheeses (46.21% DM), which resulted in a significant difference (*p* = 0.0488) in comparison with SHL and DSF2 cheeses, seems to reflect the higher casein content of the corresponding milk.

In general, the cheese components with antioxidant effect were detected in major levels in cheeses from diets based on green forage than from diets based on dried forages. In particular, vitamin A showed the highest content in cheeses from FSFL and intermediate levels with FSF4 and DSF2 diets, whereas vitamin E was higher in cheeses from DSF2, FSFL and FSF4 diets. It can be noticed that with dehydrated forage the cheese content of vitamin E was almost double than that with SHL, and comparable to that with FSF4 and FSFL, in line with the content of vitamin E in the different forages (4.96, 23.63 and 22.81 mg/kg DM in SH, DSF and FSF, respectively) reported in Gannuscio et al. [28]; these results demonstrate as the dehydration process, contrarily to hay-making, did not alter the presence of vitamin E, and evidence also as the presence of vitamin E in dairy products is strictly related to the ingested amount by diet.

Total polyphenols resulted in higher levels in FSFL than in SHL cheeses, whereas intermediate contents were observed with the other diets. The same trend was not found for CT, which were higher with FSF4 diet than with DSF2 diet; however, despite the high intake and digestibility recorded with the FSF4 and FSFL diets characterized by the higher proportions of FSF, as observed in Gannuscio et al. [28], the levels detected in cheeses from all diets were very low when compared to the respective total polyphenols ingested by ewes. This aspect suggests that CT of sulla forage could be degraded at ruminal level and metabolized along the intestine, thus modified with respect to their original structure; in this way, the derived metabolites, of lower molecular weight, are carried by circulation and incorporated into milk [4,56] where are presumably detected among polyphenols. Thus, the presence of polyphenols in cheeses, due to the transfer of dietary tannic or non-tannic compounds and/or their metabolites to milk, demonstrates a certain degree of their bioaccessibility by which it is possible to exploit their antioxidant properties, in accordance with other authors [4,9,22].

The antioxidant capacity of cheeses, expressed as TEAC, was higher with all diets based on green forage and comparable between cheeses obtained using dried forages. This improvement in the antioxidant activity induced by FSF can be attributed to the synergy effect of vitamins A and E and polyphenols. Whereas the only vitamins content in DSF2 cheeses was not able to induce a marked improvement of their antioxidant capacity, although the latter was statistically comparable to that of FSFL cheeses.

The oxidative stability of cheese fat was assessed by determining POV and TBARS as indexes of primary and secondary lipid oxidation, respectively. POV was lower in cheeses manufactured with milk from ewes fed with SHL and FSFL diets; presumably, the better oxidative stability of SHL cheeses was due to their lower PUFA content, as discussed later, whereas that of FSFL cheese can be explained by their higher antioxidant protection exerted by vitamins and polyphenols. Instead, as emerged from the higher TBARS value, the secondary oxidation interested more the SHL cheeses, which were less rich in antioxidant molecules.

Marked effects of the diet were found for all colour parameters detected on the external surface of cheeses: lightness (L*) was more apparent in the cheeses obtained with FSF2 diet, while the cheeses produced with DSF2, FSF4 and FSFL diets showed higher values of yellowness (b*) and corresponding lower values of redness (a*), indicating shifts towards yellow and green colorations, respectively. These results may be attributable to the transfer in cheeses of carotenoid pigments [57], especially lutein which, according to Rufino-Moya et al. [58], is presumably present in greater amounts in the diets that contain higher levels of sulla forage, either fresh or dehydrated. The detection of internal colour on the cheese paste revealed a single difference for redness (a*), lower in cheeses produced with FSFL diet; similarly for the external surface, the lower redness of FSFL cheeses corresponded to a higher yellowness, although the latter did not differ significantly, and may also be linked to the major content of carotenoids derived from the green forage in the diet.

The greater hardness, expressed as resistance to compression, was found in cheeses obtained with FSFL diet and reflects the major consistency of their paste, justified by their lower humidity and then by their higher DM percentage, as previously evidenced.

Parity did not influence the physico-chemical parameters of cheese, with the exception of hardness, higher in cheeses from FL ewes, presumably linked to the concomitant effects of their higher DM and lower fat, although both parameters were not significantly different.

### 3.3. Cheese Fatty Acid Profile

The fatty acid composition of cheese, reported in Table 3, Table 4 and Table 5, was greatly influenced by the diet, whereas was not affected by parity.

Regarding the short and medium chain FA (Table 3), the higher levels of C6:0, C8:0, C10:0, C12:0 and C14:0 acids were recorded with the diet based on DSF, while the C16:0 was lower in the diet with exclusive FSF provided ad libitum. As known, FA with short- and medium-chain, and partially the C16:0, are synthesized ex novo in the udder tissues from acetic acid, that is their precursor formed in the rumen by the microbial fermentation of cellulose; their major presence in the DSF2 cheeses can be related to the high fiber intake of ewes fed DSF, favoured by the high dotation of dehydrated forage in NDF and cellulose characterized by a smaller encumbrance, as evidenced in Gannuscio et al. [28].

Among long-chain FA (Table 4), the stearic acid (C18:0) was not affected by the diet, whereas C20:0 and C22:0 acids were higher with the diet FSF4. It has been widely recognized that dietary saturated FA (SFA) contributes to enhancing serum cholesterol and, consequently, the risk of cardiovascular diseases (CVD) [50]. According to Santos-Silva et al. [52], the hypercholesterolemic effect is absent for the stearic acid (C18:0), low for lauric (C12:0) and palmitic (C16:0) acids, and higher for the myristic acid (C14:0). However, it has to be noticed how current evidence based on most recent data meta-analyses do not support the recommendations encouraging high consumption of n-3 FA and PUFA and suggesting to limit the saturated fat intake to prevent the risk of CVD [58,59]; particularly, high intake of SFA has been linked to a reduced risk of stroke [60], whereas no association was found between consumption of whole-fat dairy products and increased risks of CVD [61].

The cheeses obtained with SHL diet showed greater single (Table 3: C13:0 iso, C15:0 iso, C15:0 anteiso, C17:0 iso) and total amounts (Table 5) of branched-chain FA. These FA are synthesised by ruminal bacteria and their presence is conseguence of their activity, favoured by a higher forage:concentrate ratio and then by the fiber level in the diet [62]; this explains as the higher content of branched-chain FA was recorded for the SHL group that ingested highly lignified fiber [28]. The branched-chain FA showed to have cytotoxic effects on breast cancer cells [63] and, in particular, the form C15:0 iso exerts an inhibitory action inducing apoptosis in cancer cells as those of prostate cancer, leukemia and breast adenocarcinoma [64]; this antitumoral activity is considered to be equal to that recognized for CLA [62].

The oleic acid (C18:1 c9), the main of monounsaturated FA (MUFA), as well as C16:1 c9, showed higher levels with the SHL diet, whereas the other C18:1 cis FA resulted higher with FSFL diet; since oleic acid derives also from the rumen biohydrogenation of PUFA, its increase, together with that of branched chain FA, can be attributed to the low content of CT in the sulla hay that was not able to limit the activity of microflora in the rumen, as presumably occurred with the other diets containing CT from FSF.

The FSFL diet induced in cheeses the increase of PUFA (Table 4), until now interesting for their presumed health benefits, such as linoleic acid (LA, C18:2 n-6), ALA (C18:3 n-3), and RA (C18:2 c9 t11) together with the trans vaccenic acid (TVA, C18:1 t11) that is its precursor in the rumen; these changes in the FA profile of cheese fat can be attributed to the high presence of PUFA in FSF, as well as to the inhibitory effect of CT on the rumen biohydrogenation of PUFA.

The TVA is the precursor of RA, and represents the intermediate product of biohydrogenation of PUFA, especially LA and ALA, carried out by ruminal bacteria [65]. Indeed, the presence of TVA was favoured in cheeses from the diet based on exclusive FSF, due to its high content of both ALA and CT in FSF. Once again, the CT of FSF showed to inhibit partially the rumen biohydrogenation of PUFA, favouring the production of TVA. Since the RA in ruminants milk originates mainly from the enzymatic desaturation of TVA obtained by the activity of Δ9-desaturase in the udder tissues [65], the content of TVA and RA is highly related, as confirmed by the major presence of both FA in the FSFL cheeses.

A significant increase in the ALA content in cheese occurred also with DSF2, that allowed to reach a level comparable to that with FSFL (Table 4); this result was certainly due to the ALA level, which was higher in both fresh and dehydrated forage than in SH (6.24, 7.04 and 0,54 g/kg DM, respectively; [28]). However, the ALA increase in DSF2 cheese did not correspond to an increase in TVA and RA, as occurred in FSFL cheeses. This result can be attributed to the reduction of CT to which FSF was subjected during dehydration and pelleting (from 17.9 to 5.4 g delphinidin equivalent (DE)/kg DM; [28]) as a consequence of the effect of heater temperatures [66]; indeed, the lower CT content detected in DSF can explain its moderate effect in inhibiting the ruminal biohydrogenation and forming TVA, and then RA.

The RA. the main and the most abundant among the isomers of CLA, can be synthetised in the rumen for isomerization of LA, but the most part is formed by desaturation of TVA in the tissues of ruminants, then also in the udder during milk secretion; thus, RA is present mainly in the fat of meat and dairy products from ruminants. In the last years, the RA has been the subject of interest for its health properties; in particular, it is recognized for its antitumoral activity, but is retained active in the prevention of atherosclerosis, therefore against the onset of CVD, by reducing cholesterolemia (LDL, low-density lipoprotein) and the triglycerides plasma levels, and shows also immunomodulatory and antidiabetic functions, reduces oxidative stress, and contributes to osteogenesis and in the control of obesity [67].

Among the factors affecting the RA content in milk, the feeding of dairy animals is determinant since it provides the precursors (LA and ALA) from which it derives; accordingly, feeding based on fresh forage with high dotation of PUFA, in particular ALA, is known to be responsible for the major RA enrichment of dairy products [68,69]. However, since FSF is rich in ALA as well as in CT, this explains because the FSFL diet, in which the forage component is exclusively represented by FSF, was associated to the higher level of TVA and RA in cheese, other than to the marked increase of total PUFA; indeed, the inhibiting action of PUFA biohydrogenation exerted by CT allowed the PUFA passage and absorption in the intestine, and then the transfer in milk and cheese.

Mainly, the exclusive presence of FSF in the diet improved the FA profile of cheeses in relation to health quality, at least until the current guidelines will be reassessed in accordance with the new scientific evidence about the effective role of SFA and PUFA intake in favouring or preventing the risks of CVD [61]. In this investigation, the FSFL diet contributed to reducing SFA and raising PUFA, thus determining a marked improvement of PUFA/SFA and n-6/n-3 ratios, as well as of the estimated health indexes (Table 5). In particular, the PUFA/SFA ratio of FSFL cheeses increased towards the threshold of 0.45 recommended for foods to control the level of serum cholesterol to preserve human health [70].

Instead, DSF determined a slight increase in SFA, for the contribution of C10:0 that has no effect on human health, and implied also a reduction in MUFA, due to the reduction of oleic acid; moreover, DSF induced a level of PUFA lower than that with FSFL diet and analogous to those of other diets, but greatly reduced the n-6/n-3 ratio and the corresponding ratio based on their precursors, LA/ALA, for the effect of the high level of total n-3 FA, especially ALA. However, with diets based on DSF and FSF, the level of n-6/n-3 ratio was always below the threshold (<5) recommended by FAO/WHO [71] in the human diet to prevent and treat chronic diseases, whereas this threshold was highly exceeded with the SHL diet (6.35).

With regard to the other estimated health indexes (Table 5), the FSFL diet favoured the reduction of the thrombogenic index, and the raising of the Health Promoting Index, while the h/H index was maximum with SHL diet, the latter due to the high content of oleic acid in the corresponding cheeses. The DSF worsened the Health Promoting Index of cheese fat, which reached the minimum level, but induced values of thrombogenic and h/H indexes which were similar to those of FSF2 and FSF4 cheeses.

The GHIC showed a progressively increasing trend passing from diets based on dried forages to diets with fresh forages, with the maximum value recorded in cheeses from ewes fed FSF ad libitum. This trend is in accordance with Giorgio et al. [53] who proposed the GHIC that, associating the contribution of FA (RA, total PUFA and n-3 PUFA), polyphenols and antioxidant capacity (TEAC), is able to score specifically the health-promoting value of cheeses obtained from animals fed pasture or fresh forage. The GHIC values of cheeses were above the range (16–23) recorded for sheep cheeses by Di Trana et al. [72], with exception of that of SHL cheeses, confirming the fundamental role of fresh forage-based diets to enhance the health properties of cheeses.

Finally, in FSFL cheeses, the Δ-9 desaturase indexes of C14:1 and C16:1 were lower with slight differences, whereas the index related to TVA desaturation to form RA was markedly lower, and presumably associated to the higher content in TVA.

### 3.4. Cheese Biomarkers of Animals’ Feeding Regime

The components of cheeses, as well as their groupings or ratios which resulted significantly affected by the diet, were tested for their ability to differentiate cheese for the feeding regime of animals. Accordingly, linear and quadratic regressions were performed using data of fresh forage intake (% diet DM), reported by Gannuscio et al. [28], and the cheese traits from SHL, FSF2, FSF4, FSFL experimental groups and farming ewes fed pasture-based diet. Table 6 reports the relationships of those 7 parameters of cheese better related (R^2^ > 0.50) to ewes’ fresh forage intake. Among these traits, which are all comprised within the lipid fraction of cheese, the highest R^2^ coefficients were obtained with the quadratic regressions of ALA (0.9435) and LA/ALA ratio (0.9213), represented in Figure 1.

The ability of these 7 cheese components to be used as biomarkers of animals’ diet was explored developing their box plots, which allow to display their range in relation to the diets (Figure 2).

Observing the box plots, it is possible to appreciate as the levels of ALA and LA/ALA ratio were more effective in distinguishing cheeses in relation to animals’ feeding regime. In particular, the levels of ALA in cheeses from SHL diet without fresh forage showed overlaps only with the levels of cheeses from FSF2 diet, whereas the levels emerged with higher fresh forage intake were above. The same trend can be observed for n-3 PUFA, but not for total PUFA and C17:0 anteiso.

An analogous effective discriminating ability emerged for the ratio LA/ALA, confirming also for this trait the robustness of the quadratic regression. The same potential was not detected for the n-6/n3 and PUFA/SFA ratios, whose levels in SHL cheeses showed no overlap only with the products obtained from ewes fed exclusively at pasture, corresponding to 100% of fresh forage intake.

Thus, both ALA and LA/ALA ratios have revealed a promising role as biomarker of cheeses produced from milk of animals fed fresh forage-based diets. Whereas also Maniaci et al. [26] reported ALA among the promising biomarkers for traceability of cheese production season, Segato et al. [25] did not find ALA among the predictors of the cheese production system.

Finally, it can be noticed also that a complete separation of cheeses obtained from DSF2 due to the levels of ALA and LA/ALA ratio occurred only with the cheeses from pasture. This circumstance represents further confirmation that dehydration may contribute to preserving the FA profile of green forage.

## 4. Conclusions

In this research, the effects of pelleting dehydrated sulla forage in the diet on dairy production of ewes was evaluated by comparison with fresh sulla forage and sulla hay. The results of this study showed that the dietary use of fresh or dehydrated sulla forage in dairy ewes feeding was able to enhance the health properties of cheeses due to the transfer of antioxidant molecules and PUFA from forage to milk.

The diet FSFL with exclusive fresh sulla forage offered ad libitum, corresponding to diet of grazing animals, was confirmed to greatly improve the health properties of cheese. Indeed, FSFL diet was responsible for increased amounts of antioxidant molecules, as vitamin A, vitamin E and polyphenols, that induced a better oxidative stability, and enhanced the level of PUFA interesting for their health properties, such as RA and ALA.

In cheeses from the DSF2 diet, with dehydrated forage, the contents of vitamin A, vitamin E and PUFA were higher than in cheeses obtained using sulla hay as exclusive forage source in the diet, and comparable to those of cheeses from the diet with 4 kg/day of FSF, whereas ALA was detected at the same high level of FSFL cheeses, leading to a strong reduction of the n-6/n-3 ratio.

On this basis, dehydration can represent a valid opportunity to preserve the potential of fresh sulla forage also in relation to the nutritional and health properties of dairy products. Thus, the use of dehydrated sulla pellets as an alternative to hay in periods of limited pasture resources can contribute to enhancing the antioxidant capacity and the FA profile of milk and cheese, with nutritional and health benefits for consumers. Moreover, ALA and the LA/ALA ratio exhibited an effective ability to differentiate the cheese from animals fed with fresh forage-based diets; accordingly, they may be proposed as molecular biomarkers to trace dairy products for the feeding system of dairy animals.

## Figures and Tables

**Figure 1 animals-12-02462-f001:**
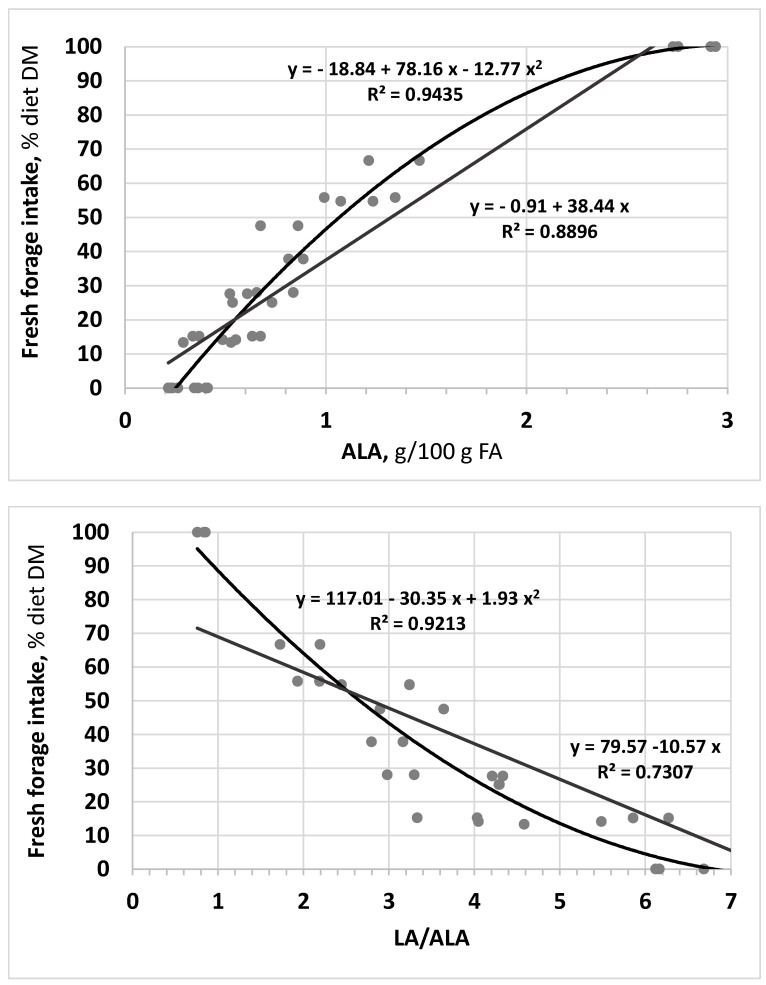
Linear and quadratic regressions of fresh forage intake (Y = % diet DM) to the level of α-linolenic acid (ALA) and the ratio linoleic acid/α-linolenic acid (LA/ALA) in cheese (X).

**Figure 2 animals-12-02462-f002:**
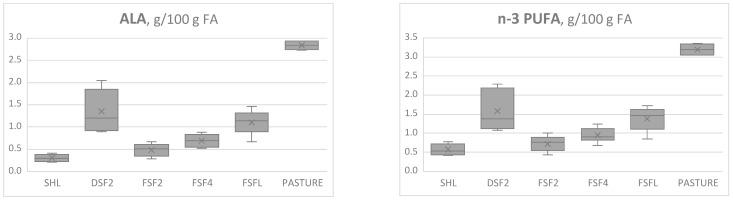
Box plots displaying the range of the potential cheese biomarkers in relation to the different diets, to explore their ability in discriminating animal’s feeding regime. SHL = sulla hay (SH) ad libitum. DSF2 = 2 kg/day dehydrated sulla forage (DSF) plus SH ad libitum. FSF2 = 2 kg/day fresh sulla forage (FSF) plus SH ad libitum. FSF4 = 4 kg/day FSF plus SH ad libitum. FSFL = FSF ad libitum. ALA = α-linolenic acid. PUFA = Polyunsaturated fatty acids. LA = linoleic acid. SFA = saturated fatty acids.

**Table 1 animals-12-02462-t001:** The effect of diet and parity on production, composition and coagulation properties of individual milk used for micro cheese-making.

	Diet (D)						Parity (*p*)			Significance *p* < (1)	
	SHL	DSF2	FSF2	FSF4	FSFL	SEM	TL	FL	SEM	D	*p*
Milk yield, g/day	1649 b	1822 a	1669 b	1663 b	1845 a	82.52	1954	1505	98.48	0.0055	0.0057
Lactose, %	4.55	4.51	4.55	4.53	4.58	0.052	4.59	4.50	0.048	0.8514	0.2020
Fat, %	5.12	5.15	5.02	4.96	4.63	0.21	4.90	5.05	0.17	0.3246	0.5318
Protein, %	4.96	5.05	4.91	5.09	5.28	0.12	4.84	5.27	0.12	0.0588	0.0229
Casein, %	3.79 b	3.93 ab	3.75 b	3.87 ab	4.03 a	0.10	3.72	4.03	0.11	0.0384	0.0598
Whey protein, %	0.90	0.90	0.90	0.98	1.00	0.037	0.87	1.00	0.029	0.0916	0.0082
Non-protein nitrogen (NPN), %	0.042	0.036	0.040	0.040	0.042	0.002	0.041	0.039	0.001	0.1906	0.2858
Urea, mg/dL	27.05	28.50	33.98	30.15	27.77	1.99	31.56	27.42	1.61	0.0875	0.0882
Somatic cells count, *n* × 1000/mL	164.19 ab	62.21 b	107.37 b	302.20 a	93.18 b	41.35	176.70	114.96	38.00	0.0032	0.2699
Somatic cells count, log_10_ n/mL	5.02 b	4.95 b	5.03 ab	5.31 a	4.86 b	0.078	5.07	4.99	0.069	0.0054	0.4166
Total bacterial count, ufc × 1000/mL	321.31	179.72	286.62	187.31	480.47	94.91	282.25	299.92	68.01	0.2297	0.8585
Total bacterial count, log_10_ ufc/mL	5.34	5.25	5.29	5.16	5.35	0.11	5.27	5.28	0.085	0.7215	0.9071
pH	6.48 ab	6.44 b	6.46 ab	6.52 ab	6.55 a	0.024	6.50	6.48	0.022	0.0136	0.4208
Titratable acidity, °SH/50 mL	4.98	5.26	5.27	5.08	5.10	0.18	4.92	5.36	0.14	0.7793	0.0445
Coagulation time (r), min	17.44	18.61	17.64	17.91	18.34	1.30	18.74	17.24	1.14	0.9048	0.3733
Curd firming time (k_20_), min	1.89	1.73	1.44	1.68	1.83	0.19	1.94	1.48	0.14	0.4498	0.0385
Curd firmness (a_30_), mm	51.91	53.73	56.26	52.61	52.14	3.26	50.15	56.51	2.62	0.8326	0.1114
Curd firmness (a_2r_), mm	57.87 c	62.24 a	61.51 a	58.33 bc	60.47 ab	0.97	58.66	61.51	0.96	0.0199	0.0584

SHL = sulla hay (SH) ad libitum. DSF2 = 2 kg/day dehydrated sulla forage (DSF) plus SH ad libitum. FSF2 = 2 kg/day fresh sulla forage (FSF) plus SH ad libitum. FSF4 = 4 kg/day FSF plus SH ad libitum. FSFL = FSF ad libitum. TL = third lambing ewes. PR = first lambing ewes. SEM = standard error of mean. (1) On the row: a, b, c = *p* < 0.05.

**Table 2 animals-12-02462-t002:** The effect of diet and parity on cheese yield and physico-chemical traits.

	Diet (D)						Parity (*p*)			Significance *p* < (1)	
	SHL	DSF2	FSF2	FSF4	FSFL	SEM	TL	FL	SEM	D	*p*
Cheese weight at 48 h, g	216.32	230.18	226.34	225.32	219.85	8.71	239.70	207.51	8.91	0.7327	0.0220
Cheese yield at 48 h, g/100 g milk	16.88	16.45	16.82	16.87	15.45	0.44	16.22	16.77	0.38	0.0522	0.3251
Dry matter (DM), %	54.00	53.70	51.85	51.16	54.72	1.45	52.40	53.77	1.19	0.2474	0.4294
Fat, % DM	47.82	48.22	47.10	47.91	47.12	1.05	48.06	47.21	0.95	0.8840	0.5366
Protein, % DM	43.53 b	43.39 b	44.75 ab	44.66 ab	46.21 a	0.83	43.90	45.11	0.79	0.0488	0.2970
Ash, % DM	5.40	4.95	5.18	5.47	5.44	0.16	5.28	5.30	0.11	0.1587	0.8756
Vitamin A, mg/kg DM	3.98 c	5.02 b	4.44 bc	4.99 b	5.96 a	0.33	5.14	4.62	0.39	<0.0001	0.3510
Vitamin E, mg/kg DM	2.97 b	5.56 a	2.74 b	5.26 a	5.09 a	0.54	4.61	4.04	0.60	<0.0001	0.5000
Condensed tannins, g DE/kg DM	0.86 ab	0.79 b	0.87 ab	0.96 a	0.86 ab	0.026	0.88	0.85	0.017	0.0012	0.2170
Polyphenols, g GAE/kg DM	6.39 b	6.61 ab	7.12 ab	7.22 ab	7.40 a	0.25	6.94	6.96	0.14	0.0050	0.8985
TEAC, mmol/kg DM	30.27 c	31.20 bc	36.02 a	38.62 a	34.77 ab	1.66	35.20	33.15	1.40	0.0001	0.3024
Peroxide value, mEq O_2_/kg fat	0.33 b	0.62 a	0.74 a	0.66 a	0.42 b	0.077	0.49	0.62	0.084	<0.0001	0.2854
TBARS, mg MDA/kg DM	0.36 a	0.27 b	0.25 b	0.22 b	0.23 b	0.058	0.25	0.28	0.080	<0.0001	0.7346
Internal colour											
Lightness, L*	87.36	87.46	88.56	87.05	86.13	1.53	88.11	86.51	1.30	0.7493	0.3893
Redness, a*	−3.20 ab	−3.82 bc	−2.93 a	−3.49 abc	−4.24 c	0.26	−3.44	−3.63	0.26	<0.0001	0.6199
Yellowness, b*	13.16	13.69	12.68	13.68	15.19	0.94	13.26	14.14	0.80	0.1905	0.4411
External colour											
Lightness, L*	89.66 ab	89.24 b	91.32 a	89.65 ab	89.02 b	0.59	90.50	89.06	0.67	<0.0001	0.1339
Redness, a*	−1.62 b	−2.50 c	−1.09 a	−2.21 c	−2.12 c	0.20	−1.66	−2.15	0.24	<0.0001	0.1476
Yellowness, b*	9.04 a	10.30 a	7.20 b	9.71 a	9.67 a	0.57	8.43	9.93	0.64	<0.0001	0.1092
Hardness, N/mm^2^	0.74 b	0.76 b	0.81 ab	0.77 ab	0.90 a	0.037	0.74	0.86	0.030	0.0097	0.0074

SHL = sulla hay (SH) ad libitum. DSF2 = 2 kg/day dehydrated sulla forage (DSF) plus SH ad libitum. FSF2 = 2 kg/day fresh sulla forage (FSF) plus SH ad libitum. FSF4 = 4 kg/day FSF plus SH ad libitum. FSFL = FSF ad libitum. TL = third lambing ewes. FL = first lambing ewes. DM, dry matter. DE, delphinidin equivalent. GAE, gallic acid equivalent. TEAC = trolox equivalent antioxidant capacity. TBARS, thiobarbituric acid–reactive substances. MDA, malonylaldehyde. SEM = standard error of mean. (1) On the row: a, b, c = *p* < 0.05.

**Table 3 animals-12-02462-t003:** The effect of diet and parity on fatty acid (FA) composition (g/100 g FA) of cheese fat: short and medium chain FA.

	Diet (D)						Parity (*p*)			Significance *p* < (1)	
	SHL	DSF2	FSF2	FSF4	FSFL	SEM	TL	FL	SEM	D	*p*
C4:0	2.46	2.43	2.50	2.60	2.41	0.10	2.61	2.36	0.068	0.6780	0.0188
C6:0	2.53 b	2.91 a	2.71 ab	2.67 ab	2.77 ab	0.074	2.77	2.66	0.051	0.0267	0.1450
C7:0	0.018	0.030	0.022	0.027	0.035	0.007	0.021	0.032	0.005	0.5113	0.1488
C8:0	2.44 b	3.06 a	2.70 ab	2.60 b	2.83 ab	0.10	2.71	2.74	0.085	0.0042	0.8252
C9:0	0.041	0.057	0.054	0.052	0.073	0.010	0.046	0.064	0.007	0.2802	0.0875
C10:0	6.79 c	9.04 a	7.88 abc	7.31 bc	8.20 ab	0.39	7.59	8.10	0.34	0.0028	0.3073
C11:0	0.32	0.39	0.35	0.34	0.28	0.028	0.33	0.34	0.023	0.1204	0.6614
C12:0	3.82 b	4.89 a	4.35 ab	4.08 ab	4.61 ab	0.24	4.20	4.51	0.22	0.0132	0.3440
C13:0 *iso*	0.034 a	0.010 b	0.019 ab	0.013 b	0.004 b	0.005	0.016	0.016	0.003	0.0032	0.9009
C13:0 *anteiso*	0.049	0.052	0.050	0.051	0.039	0.005	0.046	0.050	0.004	0.1425	0.4999
C13:0	0.13	0.16	0.16	0.14	0.16	0.016	0.14	0.16	0.013	0.7360	0.1992
C14:0 *iso*	0.16	0.15	0.14	0.16	0.12	0.012	0.14	0.15	0.009	0.1159	0.4176
C14:0	11.08 b	12.28 a	11.35 ab	11.04 b	10.80 b	0.26	11.48	11.14	0.22	0.0038	0.3108
C15:0 *iso*	0.32 a	0.24 b	0.26 b	0.24 b	0.17 c	0.011	0.25	0.24	0.008	<0.0001	0.7338
C15:0 *anteiso*	0.49 a	0.39 bc	0.47 ab	0.46 ab	0.35 c	0.024	0.42	0.44	0.020	0.0004	0.5105
C14:1 *c9*	0.23	0.22	0.21	0.21	0.17	0.020	0.21	0.21	0.019	0.1039	0.9120
C15:0	1.01	1.07	1.06	1.05	0.95	0.047	1.01	1.05	0.035	0.4427	0.4117
C16:0 *iso*	0.35	0.35	0.34	0.34	0.32	0.015	0.33	0.35	0.013	0.5364	0.3126
C16:0	28.45 a	27.96 a	27.71 a	27.51 a	25.01 b	0.56	27.58	27.07	0.57	<0.0001	0.5339
C17:0 *iso*	0.62 a	0.47 b	0.56 ab	0.58 ab	0.59 ab	0.029	0.59	0.55	0.021	0.0189	0.2267
C16:1 *t9*	0.041 b	0.051 ab	0.054 ab	0.069 ab	0.094 a	0.012	0.076	0.047	0.011	0.0092	0.0721
C17:0 *anteiso*	0.23 c	0.24 bc	0.22 c	0.27 ab	0.31 a	0.010	0.052	0.26	0.008	<0.0001	0.7058
C16:1 *c9*	1.31 a	1.13 bc	1.17 ab	1.24 ab	1.01 c	0.050	1.12	1.22	0.053	0.0002	0.2130
C17:0	0.71	0.71	0.70	0.71	0.66	0.018	0.68	0.72	0.014	0.2219	0.0630
C17:1 *c9*	0.19	0.19	0.19	0.18	0.18	0.008	0.17	0.19	0.008	0.2774	0.0773

SHL = sulla hay (SH) ad libitum. DSF2 = 2 kg/day dehydrated sulla forage (DSF) plus SH ad libitum. FSF2 = 2 kg/day fresh sulla forage (FSF) plus SH ad libitum. FSF4 = 4 kg/day FSF plus SH ad libitum. FSFL = FSF ad libitum. TL = third lambing ewes. FL = first lambing ewes. SEM = standard error of mean. (1) On the row: a, b, c = *p* < 0.05.

**Table 4 animals-12-02462-t004:** The effect of diet and parity on fatty acid (FA) composition (g/100 g FA) of cheese fat: long chain FA.

	Diet (D)						Parity (*p*)			Significance *p*< (1)	
	SHL	DSF2	FSF2	FSF4	FSFL	SEM	TL	FL	SEM	D	*p*
C18:0 *iso*	0.047	0.038	0.038	0.036	0.067	0.009	0.034	0.056	0.005	0.1162	0.0122
C18:0	7.97	6.26	7.09	7.62	6.36	0.61	7.34	6.78	0.50	0.1576	0.4448
C18:1 *t11*, TVA	1.51 b	1.29 b	1.90 b	1.95 b	3.44 a	0.28	1.88	2.16	0.18	0.0006	0.2867
C18:1 *c9*	17.29 a	13.20 bc	15.00 ab	15.28 ab	12.70 c	0.61	14.88	14.50	0.54	0.0006	0.6231
C19:0	0.010 b	0.014 b	0.009 b	0.028 ab	0.094 a	0.015	0.021	0.041	0.009	0.0051	0.1552
C18:2 n 6, LA	2.31 ab	2.14 b	2.34 ab	2.50 ab	2.69 a	0.10	2.39	2.40	0.079	0.0087	0.9481
C20:0	0.34 ab	0.27 b	0.29 b	0.40 a	0.40 a	0.027	0.34	0.34	0.026	0.0012	0.8365
C18:3 n-6,	0.031	0.026	0.033	0.019	0.031	0.006	0.024	0.031	0.005	0.4680	0.3428
C20:1 *c9*	0.060 a	0.036 b	0.059 a	0.057 ab	0.073 a	0.005	0.055	0.059	0.003	0.0035	0.4238
C18:3 n-3, ALA	0.30 c	1.35 a	0.57 b	0.59 b	1.15 a	0.068	0.78	0.80	0.060	<0.0001	0.8460
C18:2 *c9 t11* CLA, RA	0.82 ab	0.72 b	0.93 ab	0.92 ab	1.12 a	0.086	0.80	0.95	0.066	0.0329	0.7775
C20:2 n-6	0.003 b	0.005 b	0.004 b	0.003 b	0.017 a	0.004	0.003	0.009	0.002	0.0450	0.1153
C22:0	0.15 ab	0.13 b	0.13 b	0.17 a	0.15 ab	0.011	0.14	0.14	0.011	0.0063	0.9847
C20:3 n-6	0.027	0.012	0.016	0.011	0.011	0.004	0.011	0.020	0.003	0.0564	0.0365
C20:4 n-6, AA	0.11	0.094	0.11	0.11	0.11	0.006	0.097	0.11	0.005	0.2775	0.0220
C20:5 n-3, EPA	0.10	0.11	0.11	0.11	0.13	0.006	0.11	0.12	0.005	0.1194	0.2291
C22:5 n-3, DPA	0.11	0.095	0.11	0.11	0.10	0.008	0.10	0.11	0.006	0.3268	0.3953
C22:6 n-3, DHA	0.012	0.006	0.015	0.013	0.030	0.008	0.016	0.014	0.006	0.3485	0.7662
Other C16:1 *cis*	0.020 b	0.058 ab	0.016 b	0.038 ab	0.072 a	0.013	0.043	0.038	0.009	0.0130	0.7076
Other C18:1 *trans*	2.44 ab	1.73 b	2.46 ab	2.49 ab	3.61 a	0.38	2.34	2.76	0.28	0.0411	0.3064
Other C18:1 *cis*	1.63 c	2.11 abc	1.97 bc	2.25 ab	2.70 a	0.14	2.20	2.06	014	0.0001	0.4601
Other C18:2 n-6	0.95 b	1.24 b	0.89 b	1.22 b	2.12 a	0.17	1.14	1.43	0.11	0.0010	0.0791
Other CLA isomers	0.15 b	0.21 ab	0.16 b	0.17 b	0.29 a	0.022	0.19	0.20	0.015	0.0014	0.7156
Total CLA isomers	0.97 b	0.94 b	1.09 ab	1.09 ab	1.40 a	0.10	1.08	1.12	0.074	0.0274	0.7463

SHL = sulla hay (SH) ad libitum. DSF2 = 2 kg/day dehydrated sulla forage (DSF) plus SH ad libitum. FSF2 = 2 kg/day fresh sulla forage (FSF) plus SH ad libitum. FSF4 = 4 kg/day FSF plus SH ad libitum. FSFL = FSF ad libitum. TL = third lambing ewes. FL = first lambing ewes. TVA = trans vaccenic acid. LA = linoleic acid. ALA = α-linolenic acid. CLA = conjugated linoleic acid. RA = rumenic acid. AA = arachidonic acid. EPA = eicosapentaenoic acid. DPA = docosapentaenoic acid. DHA = docosahexaenoic acid. SEM = standard error of mean. (1) On the row: a, b, c = *p* < 0.05.

**Table 5 animals-12-02462-t005:** The effect of diet and parity on fatty acid (FA) profile (g/100 g FA) and health indexes of cheese fat.

	Diet (D)						Parity (*p*)			Significance *p* < (1)	
	SHL	DSF2	FSF2	FSF4	FSFL	SEM	TL	FL	SEM	D	*p*
Total FA, % DM	43.59	44.96	44.15	44.90	45.23	0.79	44.96	44.17	0.78	0.2132	0.4838
Non identified FA	0.13 b	0.33 a	0.14 b	0.19 ab	0.30 ab	0.050	0.17	0.26	0.032	0.0382	0.0601
Branched chain FA	2.31 a	1.94 b	2.10 ab	2.16 ab	1.97 b	0.064	2.08	2.12	0.054	0.0014	0.6150
Saturated FA, SFA	70.50 b	73.73 a	71.15 ab	70.54 bc	67.69 c	0.87	71.08	70.36	0.77	0.0016	0.5164
Monounsaturated FA	24.52 a	19.99 b	23.16 a	23.67 a	24.16 a	0.63	22.98	23.21	0.60	0.0002	0.7876
Polyunsaturated FA, PUFA	4.93 c	6.04 b	5.24 bc	5.85 bc	7.74 a	0.37	5.76	6.16	0.26	0.0005	0.2872
Unsaturated FA, UFA	29.34 ab	25.96 c	28.69 bc	29.29 ab	32.02 a	0.84	28.75	29.38	0.77	0.0009	0.5700
PUFA/SFA	0.070 b	0.082 b	0.074 b	0.084 b	0.12 a	0.006	0.082	0.089	0.005	0.0009	0.2640
UFA/SFA	0.42 ab	0.35 c	0.40 bc	0.42 ab	0.47 a	0.018	0.41	0.42	0.015	0.0015	0.4860
n-6 PUFA	3.43 b	3.54 b	3.41 b	3.84 b	4.97 a	0.25	3.67	4.01	0.17	0.0017	0.1791
n-3 PUFA	0.55 c	1.57 a	0.80 bc	0.87 b	1.40 a	0.079	1.02	1.06	0.063	<0.0001	0.6675
n-6/n-3	6.37 a	2.44 d	4.99 b	4.04 c	3.74 c	0.20	4.26	4.37	0.16	<0.0001	0.6152
LA/ALA	7.94 a	1.79 d	5.12 b	3.63 c	2.55 cd	0.25	4.16	4.25	0.17	<0.0001	0.7314
Thrombogenic index (TI)	3.03 a	2.79 a	2.89 a	2.87 a	2.21 b	0.11	2.81	2.70	0.11	<0.0001	0.4851
Health Promoting Index (HPI)	0.37 ab	0.31 c	0.36 bc	0.38 ab	0.42 a	0.015	0.36	0.38	0.014	0.0005	0.3409
h/H	0.51 a	0.43 b	0.47 ab	0.49 a	0.47 ab	0.017	0.47	0.48	0018	0.0089	0.8557
GHIC	16.39 c	25.58 b	26.19 b	29.27 b	37.76 a	1.84	26.92	27.15	1.74	<0.0001	0.9277
C14:1 *c9* Δ-9 desaturase ratio	0.020 a	0.017 ab	0.019 ab	0.018 ab	0.016 b	0.001	0.018	0.018	0.001	0.0428	0.7585
C16:1 *c9* Δ-9 desaturase ratio	0.044 a	0.039 b	0.041 ab	0.043 a	0.039 b	0.002	0.039	0.043	0.002	0.0476	0.1495
C18:1 *c9* Δ-9 desaturase ratio	0.68	0.68	0.69	0.67	0.68	0.017	0.67	0.69	0.015	0.9682	0.3418
RA Δ-9 desaturase ratio	0.35 a	0.37 a	0.33 a	0.34 a	0.25 b	0.021	0.33	0.32	0.018	0.0025	0.8372

SHL = sulla hay (SH) ad libitum. DSF2 = 2 kg/day dehydrated sulla forage (DSF) plus SH ad libitum. FSF2 = 2 kg/day fresh sulla forage (FSF) plus SH ad libitum. FSF4 = 4 kg/day FSF plus SH ad libitum. FSFL = FSF ad libitum. TL = third lambing ewes. FL = first lambing ewes. LA = linoleic acid. ALA = α-linolenic acid. Thrombogenic index = (C14:0 + C16:0 + C18:0)/(0.5 × MUFA + 0.5 × n-6 PUFA + 3 × n-3 PUFA + n-3/n-6) (Ulbricht and Southgate, 1991). Health-promoting index = (n-3 PUFA + n-6 PUFA + MUFA)/(C12:0 + 4 × C14:0 + C16:0) (Chen et al., 2004). h/H = hypocholesterolemic FA to hypercholesterolemic FA ratio = (C18:1 c9 + C18:2 n-6 + C20:4 n-6 + C18:3 n-3 + C20:5 n-3 + C22:5 n-3 + C22:6 n-3)/(C14:0 + 16:0) (Santos-Silva et al., 2002). GHIC = general health index of cheese (Giorgio et al., 2019). C14:1 c9 Δ-9 desaturase ratio = C14:1 c9/C14:1 c9 +C14:0. C16:1 c9 Δ-9 desaturase ratio = C16:1 c9/C16:1 c9 + C16:0. C18:1 c9 Δ-9 desaturase ratio = C18:1 c9/C18:1 c9 + C18:0. RA Δ-9 desaturase ratio = rumenic acid (RA)/trans vaccenic acid (TVA) + RA. SEM = standard error of mean. (1) On the row: a, b, c, d = *p* < 0.05.

**Table 6 animals-12-02462-t006:** Significant (*p* < 0.0001) linear (Y = a + bX) and quadratic (Y = a + bX + cX2) regressions of fresh forage intake (Y, % diet DM) to the level of potential biomarkers of cheese (X) to discriminate animal’s feeding regime (R2 > 0.50).

	X, Biomarker		Intercept	Slope		R^2^
			a	b	c	
linear	α-linolenic acid, ALA	g/100 FA	−0.91	38.45		0.8896
	n-3 PUFA	g/100 FA	−8.33	35.96		0.8697
	C17:0 *anteiso*	g/100 FA	−88.07	445.75		0.7482
	LA/ALA		79.57	−10.57		0.7307
	PUFA	g/100 FA	−58.04	14.49		0.6895
	n-6/n-3		98.17	−14.66		0.6679
	PUFA/SFA		−43.62	845.13		0.6098
quadratic	α-linolenic acid, ALA	g/100 FA	−18.84	78.16	−12.77	0.9435
	n-3 PUFA	g/100 FA	−28.63	68.61	−8.85	0.9043
	C17:0 *anteiso*	g/100 FA	5.09	−227.05	1158.35	0.7701
	LA/ALA		117.01	−30.35	1.93	0.9213
	PUFA	g/100 FA	−120.74	33.32	−1.31	0.7031
	n-6/n-3		148.10	−40.20	2.80	0.7773
	PUFA/SFA		−111.48	2251.82	−6615.17	0.6382

PUFA = Polyunsaturated fatty acids. LA = linoleic acid. SFA = saturated fatty acids.

## Data Availability

Not applicable.

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
