# Peer review of "Feeding Dairy Ewes with Fresh or Dehydrated Sulla (Sulla coronarium L.) Forage. 2. Effects on Cheese Enrichment in Bioactive Molecules"

_animals, 2022, doi:10.3390/ani12182462_

Round 1

Reviewer 1 Report

Dear authors, 

Thank you for the intresting research. 

I am aware that there is a huge amount of work that has been done in this research. Specially, maintaining 20 ewes for two month !!!

Some aspects of the manuscript should be considered before publication:

1. considering the research is a nutritional sutdy, diet information should be reported (raw materials, inclusion proportion). Moreover the nutrtional value of the diets used, also should be reported;

2. Statistical analysis: the observation used be defined clearly, daily, period/animal observation. State the interactions checked before not including them. 

Once again, thank you very much for the effort made in this research. 

Reviewer 2 Report

The manuscript submitted by Ponte et al., deals with the evaluation of the effects induced on cheese quality after feeding ewes with fresh or dehydrated sulla forage.

The paper is quite interesting given the richness in polyphenols and PUFA of Sulla coronarium, and as a consequence the possibility of obtaining a qualitative improvement of dairy products.

The language is clear and appropriate, so no particular changes are required with regard to the style.

Specific comments:

-          In my opinion the simple summary should be shortened.

-          L67 and in the rest of the manuscript: check how bibliographic references are indicated. Stay consistent to the style of the Journal (for example, [1] or [2,3], or [4–6]).

-          L113-122: the objectives of the study are clearly described, however the hypothesis on which the study was defined and organized must be reported.

-          L128: milk yield?

-          Paragraph 2.3 contains all the information necessary to understand and reproduce the applied methods, however I believe it is better to divide this section into subparaphrams to facilitate the understanding by the reader. An example could be to divide the evaluations conducted on milk and cheese, or to separate the physical analyzes from the chemical ones.

-          Given the amount of information reported in the paper, I would invite the authors to separate the Results section from the Discussion. I am of the opinion that this will also enhance the scientific impact of the work.

Reviewer 3 Report

In general, the work is well written and well formatted. This work does compliment the work done in the first part of the experiment that evaluated the effect of sulla on milk properties and animal performance and nutrient utilization . The are a few changes requested in the format of the paper but in general the paper is well done. Experiments such as this should be encouraged especially in this time where animal feed based on grains may be limited.

Review in text references to ensure it complied with journal standards (eg. line 139)

The results and discussion sections should be separate as requested in the first part of the study. This will improve the clarity of the paper to the readers.  

Line 409: revise “this” to “the”

Round 2

Reviewer 1 Report

Dear authors, 

Thank you for the revision made on the manuscripts. 

A minor change, should be made.

The Y axis of the graphics should have the titles and the units next to it, not as a graph title. Even if we understand the content of the graphics. However, it is a basic graphics layout that should be respected. 

Kind regards.

Author Response

Thank you for your suggestion. The title to the Y axis has been added